# Ti*_n_*O_2*n*−1_ Suboxide Phases in TiO_2_/C Nanocomposites Engineered by Non-hydrolytic Sol–Gel with Enhanced Electrocatalytic Properties

**DOI:** 10.3390/nano10091789

**Published:** 2020-09-09

**Authors:** Shuxian Zou, Romain Berthelot, Bruno Boury, Pierre Hubert Mutin, Nicolas Brun

**Affiliations:** ICGM, Univ Montpellier, CNRS, ENSCM, Montpellier, France; shuxian.zou@etu.unistra.fr (S.Z.); romain.berthelot@umontpellier.fr (R.B.); bruno.boury@umontpellier.fr (B.B.); hubert.mutin@umontpellier.fr (P.H.M.)

**Keywords:** nanocomposite, nanocrystal, nanostructure, non-hydrolytic sol-gel, atom-economic synthesis, carbothermal reduction, titanium suboxide, magnéli phases, electrocatalysis, ORR

## Abstract

We report a non-hydrolytic sol-gel (NHSG) route to engineer original mesoporous Ti*_n_*O_2*n*−1_@TiO_2_/C nanocomposites. The synthetic approach is straightforward, solvent-free, additive-free, and meets the challenge of atom economy, as it merely involves TiCl_4_ and THF in stoichiometric amounts. We found that these nanocomposites present enhanced electrocatalytic properties towards the oxygen reduction reaction (ORR) in 0.1 M KOH. We believe that these preliminary results will open a window of opportunity for the design of metal suboxides/carbon nanocomposites through NHSG routes.

## 1. Introduction

Titanium suboxides have received considerable attention in the last decades due to their remarkable electronic and optical properties [1,2,3]. Titanium suboxides can be classified in two main families [4,5]: (i) oxygen-deficient titanium oxides, namely, TiO_2−*x*_ (where 0.001 > *x* > 0) or black TiO_2_ [6], which present point defects such as oxygen vacancies and titanium interstitials [7]; (ii) Magnéli phases, namely, TiO_2−*x*_ (where 0.25 > *x* > 0.001), for which the off-stoichiometry is accommodated by extended defects and changes in the crystal structure. Magnéli phases, also noted as Ti*_n_*O_2*n*−1_ (where *n* varies from 4 to 37) and first described in the 1950s by Magnéli et al. [8,9], present crystallographic shear (CS) planes, and can be seen as ordered combinations of rutile TiO_2_ and corundum Ti_2_O_3_ structures [3,4]. Note that, in contrast to Magnéli phases, Ti_3_O_5_ structures [10] (Ti*_n_*O_2*n*−1_, where *n* = 3) have no CS planes. However, as they satisfy the Ti*_n_*O_2*n*−1_ formula, Ti_3_O_5_ polymorphs have often been considered as the first members of the Magnéli phases’ family. Titanium suboxides depict remarkable properties [3,11,12], amongst which are electrochemical stability, high electronic conductivity (c.a. 10.4 S cm^−1^ for Ti_4_O_7_ at room temperature [13]), high electron mobility (>0.5 cm^2^ V^−1^ s^−1^ for Ti_4_O_7_ at room temperature [1]), and low band-gap as compared with stoichiometric TiO_2_ (about 0.5 eV smaller than that of TiO_2_ anatase [14]). Thus, titanium suboxides have been widely investigated in electrocatalysis [15,16,17,18,19], photocatalysis [20,21,22,23] and energy storage devices [24,25].

Several synthesis routes have been reported in the literature for the preparation of titanium suboxides, either defective or Magnéli phases. Most of them start from stoichiometric TiO_2_ and involve oxygen scavengers such as zirconium foil [19,26] and/or reducing agents such as carbon (through carbothermal reduction) [27,28,29], CaH_2_ at low temperature (350–450 °C) [30,31], or hydrogen gas at temperatures typically higher than 950 °C [24]. Amongst them, the carbothermal reduction approach is particularly interesting. It generally implies external carbon sources, usually from organic additives, combined to the oxide by sol-gel process and yielding, after carbothermal reduction, Ti*_n_*O_2*n*−1_/C composites in simple, one-pot, bottom-up approaches [28]. The sol-gel process has been largely used for the preparation of advanced metal-oxide materials, such as TiO_2_. This process presents numerous advantages, including versatility and scalability, and can be conducted under mild condition (T ≤ 100 °C, 1 atm). Regarding titanium suboxides, a few studies have been reported on sol-gel/carbothermal reduction routes starting from commercial titanium alkoxides and organic additives, such as poly(ethyleneimine), polyethyleneglycol [28], poly(styrene-*b*-2-vinylpyridine) [32], resol [25], or glucose [29]. Following such a procedure, Portehault et al. [28] obtained single-phase Magnéli/carbon nanocomposites with both structural (Ti*_n_*O_2*n*−1_ with *n* = 8, 6, 5, 4, or 3) and textural control by tuning the Ti:C ratio, the nature of the carbon source and the calcination atmosphere (argon versus nitrogen). More recently, Huang et al. [29] prepared various Ti*_n_*O_2*n*−1_/C nanocomposites employing glucose as the carbon source and a carbothermal reduction process under vacuum. The authors demonstrated the importance of the cooling conditions which govern the predominant titanium suboxide phase obtained in fine (*n* = 4, 3, 2, or 1). Those are two good examples that demonstrate the versatility of the sol-gel process combined to carbothermal reduction to design Ti*_n_*O_2*n*−1_/C nanocomposites. Although innovative and efficient, most of these approaches rely on relatively expensive precursors (e.g., synthetic polymers) and/or require organic additives in a large excess, making the all procedure wasteful of resources. With this aim, we propose hereafter to investigate an original one-pot, atom-economic synthesis route to prepare Ti*_n_*O_2*n*−1_/C nanocomposites using the non-hydrolytic sol-gel process (NHSG).

Unlike conventional aqueous sol-gel processes, NHSG is performed in organic medium and involves organic oxygen donors such as ethers or alcohols instead of water [33,34]. NHSG offers simple, one-step routes to produce oxide-based materials with well controlled compositions and textures, avoiding the use of expensive precursors or reactivity modifiers [33]. The main drawback of NHSG lies in the necessity to work under anhydrous conditions and at higher temperature (up to 200 °C) and pressure (up to c.a. 30 bars) than conventional aqueous sol-gel methods, which may hamper industrial applications. Nevertheless, NHSG has been established as a versatile methodology for the synthesis of mesoporous metal-oxides and mixed oxides in the absence of templating agents or costly drying procedures [35], metal oxide nanocrystals [36,37,38,39] as well as organic-inorganic hybrids [40,41] and metal-oxide/carbon nanocomposites [42,43]. Recently, we reported a solvent-free NHSG route to synthesize TiO_2_/C nanocomposites based on the reaction of simple ethers (diisopropyl ether or tetrahydrofuran, THF) in a stoichiometric amount with titanium tetrachloride [42]. In this atom-economic process, the ether acted not only as an oxygen donor, but also as the sole carbon source, yielding anatase TiO_2_ nanocrystals coated by an amorphous carbon film [42]. Herein, we further extend this original approach to the preparation of titanium suboxides, by demonstrating that Ti*_n_*O_2*n*−1_ phases can be easily generated by partial carbothermal reduction of the TiO_2_/C nanocomposites at relatively low temperature, i.e., 950 °C, under argon flow. This solvent-free, additive-free approach is straightforward, and the partial reduction of TiO_2_ into Ti*_n_*O_2*n*−1_ phases (with 2 ≤ n ≤ 6) allows engineering original mesoporous Ti*_n_*O_2*n*−1_@TiO_2_/C nanocomposites. The latter are rarely targeted as such, researchers generally focussing on pure phases, but we found that this nano-assembly presents enhanced electrocatalytic properties towards the oxygen reduction reaction (ORR) in 0.1 M KOH. We believe that these preliminary results open new opportunities for the atom-economic design of metal suboxides/carbon nanocomposites through NHSG routes.

## 2. Materials and Methods

### 2.1. Materials Synthesis

To avoid water, all manipulations were carried out in a glovebox under argon atmosphere (<10 ppm of water and O_2_). Titanium(IV) chloride 99%, (Sigma-Aldrich Chemie, Saint-Quentin Fallavier, France) was used as received. Tetrahydrofuran (THF) 99.8%, (Fisher Scientific, Hampton, NH, USA) was dried over a PureSolve MD5 solvent purification system (Innovative Technology Inc., Amesbury, MA, USA). The amount of water in THF was controlled with a Karl Fischer coulometer (SI Analytics, Mainz, Germany) before reaction (H_2_O < 5 ppm).

The synthesis route is well-established and was already reported in our previous work [42]. Typically, TiCl_4_ (12.52 mmol, 1.37 mL) and THF (25.04 mmol, 2.03 mL) were added into a 23 mL stainless steel digestion vessel with a polytetrafluoroethylene (PTFE) lining from Parr Instrument Company (Moline, IL, USA). The sealed digestion vessel was heated in an oven at 180 °C for 4 days under autogenous pressure. The gel was dried under vacuum at 120 °C for 5 h, yielding a xerogel labeled xer-TiO_2_. The resulting xerogel was pyrolyzed under argon atmosphere (50 mL min^−1^) at 850 or 950 °C for 1 to 8 h (heating rate 10 °C min^−1^) to obtain Ti*_n_*O_2*n*−1_@TiO_2_/C nanocomposites. These nanocomposites are labeled TiO_2_/C-*x*-*y*, with *x* the first digit of the plateau temperature in degrees Celsius; and *y* the duration at the plateau in hours. For example, TiO_2_/C-9-8 stands for the sample pyrolyzed at 950 °C for 8 h. For comparison, a TiO_2_ sample was obtained by calcination in air (50 mL min^−1^) of the xerogels at 500 °C for 5 h (heating rate 10 °C min^−1^). This sample is labeled TiO_2_-5-5.

### 2.2. Characterization

Thermogravimetric analyses (TGA) were carried out using a Netzsch Simultaneous Thermal Analyzer STA 409 PC Luxx (Netzsch, Selb, Germany), with a heating rate of 10 °C min^−1^ in the 20–1000 °C range, under a dry air atmosphere or under argon (50 mL min^−1^). X-Ray diffraction (XRD) patterns were obtained using a PANalytical X’Pert Pro MPD diffractometer (Malvern PANalytical Ltd, Malvern, UK), with the K*α* radiation of Cu (*λ* = 1.5418 Å) and a step size of 0.033° (2θ scale) into the 10–70° interval. N_2_-physisorption experiments were carried out at −196 °C on a Micromeritics TriStar 3000 (Micromeritics France, Merignac, France). Samples were out-gassed under vacuum at 110 °C overnight. Equivalent BET specific surface areas (SSA_BET_) were determined in the relative pressure range P/P_0_ from 0.08 to 0.25. The total pore volume (V_0.99_) was measured at P/P_0_ = 0.99. Transmission electron microscopy (TEM) images were acquired with a JEOL 1200 microscope (JEOL Europe, Croissy sur Seine, France). High resolution TEM analysis was performed on a JEOL 2200FS microscope (JEOL Europe, Croissy sur Seine, France) operated at 200 kV from the MEA platform (Université de Montpellier, Montpellier, France). This microscope is equipped with a field emission gun (FEG) and an in-column Omega-type energy filter. STEM-EDX (Scanning Transmission Electron Microscopy) mapping were performed using a probe size of 1.5 nm and X-rays measured with a silicon drift detector (30 mm^2^, JEOL) with a collection solid angle of 0.13 sr.

### 2.3. Electrochemical Properties

A standard ink was prepared by mixing 10 mg of material with 300 µL of a Nafion^®^ binder solution and 700 µL of absolute ethanol. The ink was ultrasonicated for at least 30 min for homogenization. A 5 µL aliquot was dropped onto a freshly polished glassy carbon rotating disk electrode (diameter 3 mm, electrode area 0.071 cm^2^) to prepare a catalyst thin film. The electrode was warmed to 40 °C in an oven prior to ink application in order to achieve better electrode coverage of the catalyst film. Electrochemical tests were performed in 0.1 M KOH (70 mL) in a standard three-electrode setup with an Ag/AgCl reference electrode and a Pt wire counter electrode. We used a RRDE-3A device (ALS Co. Ltd, Tokyo, Japan) to perform the rotating disk electrode (RDE) hydrodynamic measurements, a BioLogic VSP-300 potentiostat (BioLogic, Seyssinet-Pariset, France) and the EC-Lab^®^ Software (BioLogic, Seyssinet-Pariset, France). Koutecky–Levich analyses were performed as described in literature [44] (for more details, see Appendix A).

## 3. Results

### 3.1. Synthesis and Characterization

#### 3.1.1. NHSG Ether Route and Carbothermal Reduction: Mechanistic Insights

The NHSG ether route has been extensively used to prepare metal oxides, especially for the synthesis of titanium oxides [33,35,45]. In our previous works, we reported on the synthesis of TiO_2_ nanocrystals and TiO_2_/C nanocomposites from TiCl_4_ and THF [42]. This approach is based on the intermediate formation of alkoxide groups, which then condense with Ti-Cl to form oxo bridges (step (1) in Figure 1). In the case of THF and at moderate temperature (below 110 °C), the major organic by-product is 1,4-dichlorobutane. At higher reaction temperature, typically at 180 °C as used herein, secondary reactions are favoured, especially the formation and polymerization of alkenes resulting from the elimination of HCl (step (2) and (3) in Figure 1). These reactions, most probably catalyzed by the freshly produced TiO_2_ particles and HCl, can form highly cross-linked polymers, as shown in Figure 1. In a previous article [42], we showed that after pyrolysis at 750 °C under argon, we could take advantage of these highly cross-linked polymers to prepare TiO_2_/C nanocomposites (step (4) in Figure 1). Herein, we propose to investigate thermal post-treatments at higher temperatures, i.e., 850 and 950 °C, and to explore the opportunity to design Ti*_n_*O_2*n*−1_@TiO_2_/C nanocomposites through carbothermal reduction (step (5) in Figure 1).

The presence of a large amount of organic cross-linked polymers in the xerogel obtained right after NHSG and subsequent drying, labeled xer-TiO_2_, was confirmed by thermogravimetric analysis (TGA) under synthetic dry air flow, as shown by the weight loss of about 31 wt.% from 240 °C to 550 °C (Figure 2a). To assess the carbothermal reduction process, TGA was also performed under argon flow on xer-TiO_2_ (Figure 2b). As mentioned above, this xerogel contains TiO_2_, oxygen-poor cross-linked polymers, and some residual chlorine atoms bonded to carbon [42]. On the TGA curve, three main weight loss regions can be seen: (i) in the temperature range 40–200 °C, with a weight loss of about 2 wt.%; (ii) in the temperature range 280–500 °C, with a weight loss of about 14 wt.%; and (iii) in the temperature range 850–1000 °C, with a weight loss of about 2 wt.% (Figure 2b). While the first region is most probably related to the elimination of residual volatile compounds, the second region can be attributed to the carbonization of the cross-linked polymers through inter- and intramolecular reactions such as dehydrochlorination. The third region might be attributed to the partial carbothermal reduction of TiO_2_ to Ti*_n_*O_2*n*−1_.

#### 3.1.2. TiO_2_ and TiO_2_/C Nanocomposites: Characterization

TiO_2_/C nanocomposites were prepared by pyrolysis of xer-TiO_2_ at 950 °C for 1, 2 and 8 h. For comparison, xer-TiO_2_ was also pyrolyzed at 850 °C for 4 h, or calcined in air at 500 °C for 5 h. These materials were thoroughly characterized by TGA, elemental analysis, X-ray diffraction (XRD), nitrogen sorption at 77 K, and transmission electron microscopy (TEM).

The TGA curves of the samples pyrolyzed at 950 °C (TiO_2_/C-9-*y* samples) were nearly identical, whatever the duration of the pyrolytic treatment. The total weight loss is much lower than the one of xer-TiO_2_ (c.a. 6–7 wt.%, Table 1) and starts at higher temperature, above 450 °C. Interestingly, a significant weight gain was observed in the temperature range 300–500 °C for the TiO_2_/C-9-*y* samples (Figure 3). This feature can be ascribed to the re-oxidation of titanium suboxides and has already been reported for Ti*_n_*O_2*n*−1_/C nanocomposites in the same temperature range [29].

The formation of titanium suboxide in the TiO_2_/C-*x*-*y* samples was confirmed by XRD (Figure 4, Table 1). While anatase (A) was the major phase present in all the samples, several peaks attributed to Ti_3_O_5_ (monoclinic) and Ti_6_O_11_ were observed for the TiO_2_/C-9-*y* samples, showing that carbothermal reduction of TiO_2_ occurred at 950 °C. Conversely, in the case of the sample pyrolyzed at 850 °C, TiO_2_/C-8-4, a significant amount of rutile (R) was detected, as reported in our previous work [42], but no suboxide phase was identified, indicating that carbothermal reduction did not take place at this temperature. The presence of a substantial amount of anatase (A) after pyrolysis at 850 °C and 950 °C is remarkable, since the anatase to rutile crystal transition usually occurs below 700 °C [46]. As reported in literature [47], the presence of carbon layers onto coated titanium dioxyde particles can suppress the anatase to rutile phase transformation by preventing sintering and crystal growth. In our previous works [42,48], the same feature was observed for TiO_2_/C nanocomposites prepared by a similar NHSG procedure and pyrolyzed under argon atmosphere at 750 and 900 °C. Our results suggested that the carbon content, which was related to the thickness of the carbon layer onto TiO_2_ nanoparticles, strongly affects the anatase to rutile crystal transition [48]. No significant difference was observed between the patterns of the three samples of the TiO_2_/C-9-*y* series, confirming that the duration of the pyrolytic treatment did not influence the carbothermal reduction process. Note that the sole difference, observed in the 32–33° (2θ) region (Figure 4), suggests that a longer pyrolytic treatment at 950 °C might favor Ti_6_O_11_ over Ti_3_O_5_. Even though elemental analysis and TGA indicated that about 6 wt.% C remained in the TiO_2_/C-9-*y* samples, the presence of anatase indicates that the carbothermal reduction at 950 °C was incomplete, whereas in recent studies reported in the literature, either single-phase Magnéli [28,32] or predominant suboxide phases with a small amount of rutile and anatase [29] were obtained by carbothermal reduction below 1000 °C. Particularly, Huang et al. [29] obtained various titanium suboxides from Ti_4_O_7_ to TiO with a C:Ti molar ratio in the xerogel between 1.1 and 6.5. In our case, we employed a C:Ti molar ratio of 1.9 (i.e., 20 wt.% of carbon in xer-TiO_2_, Table 1), which should theoretically be sufficient to fully reduce TiO_2_. While the C:Ti molar ratio did not seem critical herein, several factors may explain the incomplete reduction of TiO_2_. This will be discussed later.

Besides structural properties, textural and morphological properties were investigated. The nitrogen sorption isotherms of xer-TiO_2_, TiO_2_-5-5, and the TiO_2_/C-*x*-*y* series are shown in Figure 5. The associated textural data are compiled in Table 1. All the samples are mesoporous, which is in good agreement with our previous works on TiO_2_-based materials obtained by NHSG ether routes [42]. According to the IUPAC classification [49], the isotherms are of type IV, corresponding to mesoporous solids with interparticle porosity. The TiO_2_/C-*x*-*y* samples, especially the ones treated at 950 °C, depict significantly higher specific surface areas and pore volumes than xer-TiO_2_ and TiO_2_-5-5. This feature suggests that the carbonaceous phase is highly porous, most probably due to the thermal degradation of the organic cross-linked polymer phase during pyrolysis.

The presence of a porous carbon phase was confirmed by TEM (Figure 6). While xer-TiO_2_ is composed of large agglomerates of TiO_2_ anatase nanocrystals with a diameter inferior to 20 nm (Figure 6a), TiO_2_/C-9-1 is composed of larger nanoparticles of ca. 50 nm embedded in a mesoporous carbon matrix (Figure 6b,c). These mesopores have a diameter of ca. 10 nm, in the range of the native TiO_2_ anatase nanocrystals. One may assume that these mesopores were generated during the thermal post-treatment, due to the migration/growth of TiO_2_ nanocrystals through the carbon matrix and the consumption of carbon atoms during the carbothermal reduction process. STEM-EDX mapping on TiO_2_/C-9-1 confirmed that TiO_2_ and/or Ti*_n_*O_2*n*−1_@TiO_2_ nanocrystals are surrounded by a carbon film.

### 3.2. Electrochemical Properties Towards the Oxygen Reduction Reaction (ORR)

To demonstrate the superior properties of the Ti*_n_*O_2*n*−1_@TiO_2_/C nanocomposites prepared herein, we proposed to assess their electrocatalytic activity towards the oxygen reduction reaction (ORR). The ORR is one of the most studied reactions in the field of electrochemistry, especially when dealing with energy storage and conversion devices [50,51]. In particular, ORR in aqueous alkaline media [52] has been widely studied and is of great interest for applications in metal-air batteries [53] and alkaline anion-exchange membrane fuel cells [54]. In aqueous alkaline media, ORR can proceed either by a direct four-electron pathway to directly reduce O_2_ into hydroxide ions (Equation (1)), or by a two-electron pathway with formation of peroxide ions as intermediate species (Equation (2)), followed by a two-electron reduction to hydroxide ions (Equation (3)) or disproportionation (Equation (4)).
(1)$$O_{2}+2H_{2}O+4e^{-}\to 4OH^{-}$$
(2)$$O_{2}+H_{2}O+2e^{-}\to HO_{2}^{-}+OH^{-}$$
(3)$$HO_{2}^{-}+H_{2}O+2e^{-}\to 3OH^{-}$$
(4)$$2HO_{2}^{-}\to 2OH^{-}+O_{2}$$

Cyclic voltammetry (CV) measurements were carried out in O_2_-saturated 0.1 M KOH solution. The CV curves obtained for TiO_2_/C-9-1 present a well-defined cathodic peak centered between −0.3 and −0.4 V vs. Ag/AgCl (Figure 7a), demonstrating the electrocatalytic activity of this material towards the ORR. In order to thoroughly compare the electrochemical properties of each material, linear sweep voltammetry (LSV) was performed and polarization curves were recorded from 0.2 to −1.0 V vs. Ag/AgCl, at a rotation rate of 1600 rpm (Figure 7b). The presence of an intermediary current plateau (between −0.4 and −0.6 V for TiO_2_/C-9-*y*; and between −0.5 and −0.7 V for TiO_2_/C-8-4, TiO_2_-5-5 and xer-TiO_2_) can be attributed to several factors, and may suggest either: (i) a mass transport limitation (e.g., difficult access of the electrolyte to the whole porous volume) [55], (ii) the occurrence of the ORR through two consecutive catalytic steps [56], and/or (iii) the presence of a second set of active sites with distant onset potentials and slow kinetics. Moreover, the absence of a clear mass transport limited current plateau at low potential (i.e., below −0.7 V vs. Ag/AgCl) suggests slow kinetics. While the shape of these polarization curves is still opened to debate, LSV curves with similar shapes were reported for TiO_2_ and defective TiO_2−*x*_ single crystals in the ORR in 0.1 M KOH [17]. Besides these considerations, significant differences were observed between the samples prepared herein. Both the onset potential (determined at the base of the first catalytic wave), the half-wave potential (E_cat/2_, determined at the half of the first catalytic current wave) and the catalytic current (determined at the first plateau region) increased from stoichiometric TiO_2_ (i.e., xer-TiO_2_ and TiO_2_-5-5) to sub-stoichiometric Ti*_n_*O_2*n*−1_ phases-containing samples (i.e., TiO_2_/C-9-1 and TiO_2_/C-9-8). Interestingly, TiO_2_/C-8-4, free of Ti-suboxide phase, depicts intermediary properties (Figure 7b, Table 2), which can be attributed to the presence of the amorphous carbon coating surrounding the (stoichiometric) TiO_2_ nanocrystals. As reported by Pei et al. [17] with TiO_2_/C nanocomposites, such enhanced ORR activity may be attributed to a superior electronic conductivity and an improved charge transfer across the TiO_2_-carbon interface.

To further investigate the electrocatalytic mechanisms involved herein, polarization curves were acquired at different rotation rates, from 600 to 2600 rpm (Figure 7c). The reciprocal current (I_O2_^−1^) was plotted against the reciprocal root of the angular rotation rate (ω^−1/2^) at various potentials from −0.3 to −0.55 V vs. Ag/AgCl, giving Koutecky–Levich (KL) plots (Figure 7d and Appendix A). These plots were drawn for TiO_2_/C-9-1 and TiO_2_/C-9-8. While the slope gives information about the electron transfer number, the y-intercept (equal to the reciprocal kinetic current) gives information about the kinetic current and the rate constant of the reaction [44]. As for kinetic currents, they are similar for TiO_2_/C-9-1 and TiO_2_/C-9-8, but lower than Pt/C, 20 wt.%, used as a benchmark (Table 2 and Appendix B, Figure A1). This result supports low rate constants for TiO_2_/C-9-1 and TiO_2_/C-9-8 as compared with Pt/C, 20 wt.%. As for electron transfer numbers, KL plots revealed a clear difference between the two Ti*_n_*O_2*n*−1_@TiO_2_/C nanocomposites. While TiO_2_/C-9-1 seems to be very selective for the two electrons process, TiO_2_/C-9-8 depicts an intermediary behavior between the two electrons process and the four electrons process (Figure 7d). While not fully understood, this difference may be attributed to enhanced electronic conductivity and charge transfer across the TiO_2_-carbon interface, due to the longer plateau duration at 950 °C during carbothermal reduction.

## 4. Discussion and Conclusions

Overall, the NHSG route allowed producing Ti*_n_*O_2*n*−1_@TiO_2_/C nanocomposites by a straightforward solvent-free, additive-free approach that meets the challenge of atom economy, as it merely involves TiCl_4_ and THF in stoichiometric amounts. Despite a theoretically sufficient C:Ti molar ratio, the carbothermal reduction, even after long treatment, was incomplete as compared with recent studies reported in literature [28,29,32]. NHSG routes usually yield highly condensed and well crystallized metal oxide particles [33], which is also true in the present case. Thus, one may assume that these native TiO_2_ anatase nanocrystals are more resistant to the carbothermal reduction, which was most probably limited here to the titanium atoms located at the very surface of the nanoparticles or to the smallest nanocrystals. Besides, the nature and the reactivity of the carbon produced by the NHSG route may also be a source of explanation of the incomplete carbothermal reduction, since it has been reported that the type of carbon plays an important role on the reducibility of TiO_2_ [58].

Anyway, the NHSG route seems promising and should be further explored. To better understand the mechanisms involved in the carbothermal reduction and to identify the key parameters, other NHSG routes, with various oxygen donors and carbon sources, should be considered to yield smaller nanocrystals and/or higher carbon contents. The use of innocuous alternatives to THF, such as polysaccharides [59], should be also investigated to make the whole procedure safer and more sustainable. Besides, alternative reduction protocols are currently considered to yield more advanced reduction reactions, such as higher temperatures, the use of hydrogen gas or the addition of oxygen scavengers. To conclude, we believe that these preliminary results will open a window of opportunity for the design of metal suboxides/carbon nanocomposites through atom-economic NHSG routes. This approach should not be limited to titanium suboxides and can easily be extended to other metal oxides.

## Figures and Tables

**Figure 1 nanomaterials-10-01789-f001:**
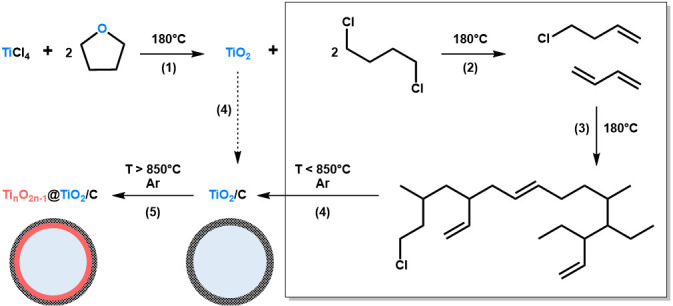
Schematic representation of the non-hydrolytic sol-gel (NHSG) route developed herein to prepare TiO_2_, TiO_2_/C and Ti*_n_*O_2*n*−1_@TiO_2_/C nanocomposites. (1) Alkoxylation/Condensation; (2) Dehydrochlorination; (3) Polymerization; (4) Carbonization; and (5) Partial carbothermal reduction. In this representation, xer-TiO_2_ corresponds to TiO_2_ together with organic by-products (box on the right side) before step (4).

**Figure 2 nanomaterials-10-01789-f002:**
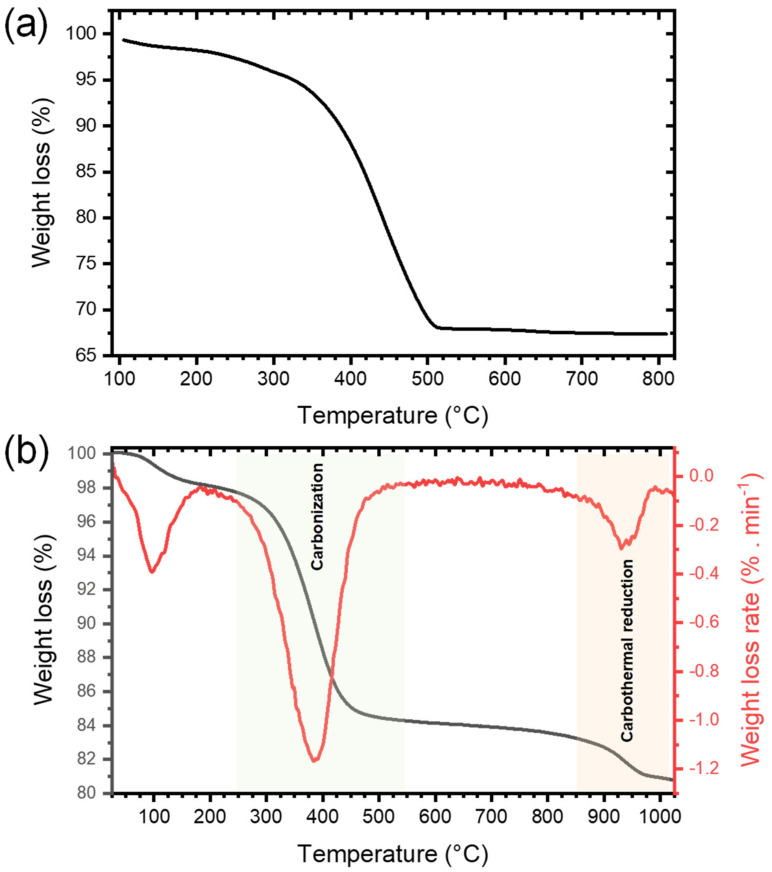
Thermogravimetric analyses performed (**a**) under synthetic dry air flow (50 mL min^−1^) and (**b**) under argon flow (50 mL min^−1^) at 10°C min^−1^ of xer-TiO_2_.

**Figure 3 nanomaterials-10-01789-f003:**
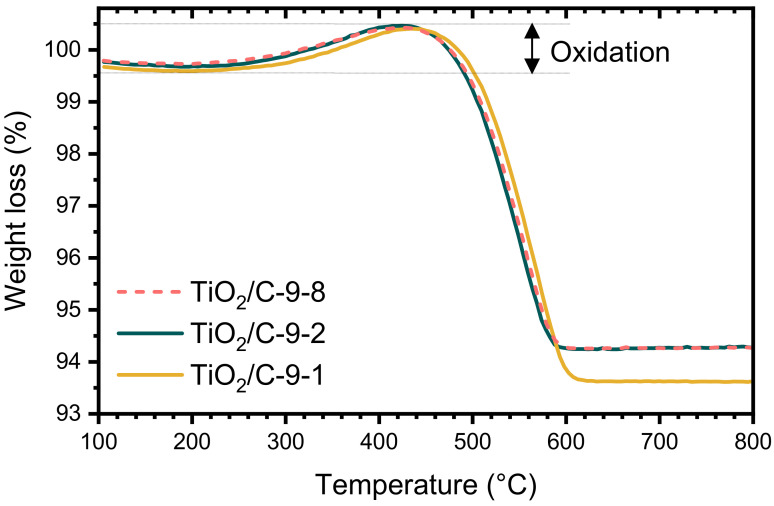
Thermogravimetric analyses (TGA) performed under synthetic dry air flow (50 mL min^−1^) at 10 °C min^−1^ of TiO_2_/C-9-*y* samples.

**Figure 4 nanomaterials-10-01789-f004:**
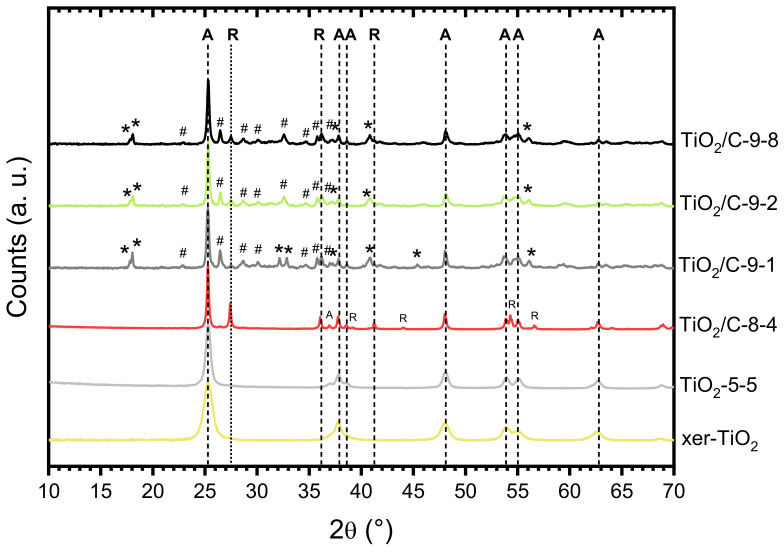
X-ray diffraction (XRD) patterns of xer-TiO_2_, TiO_2_-5-5, and the TiO_2_/C-*x*-*y* series. A: Anatase; R: Rutile; *: Ti_3_O_5_ (monoclinic); #: Ti_6_O_11._

**Figure 5 nanomaterials-10-01789-f005:**
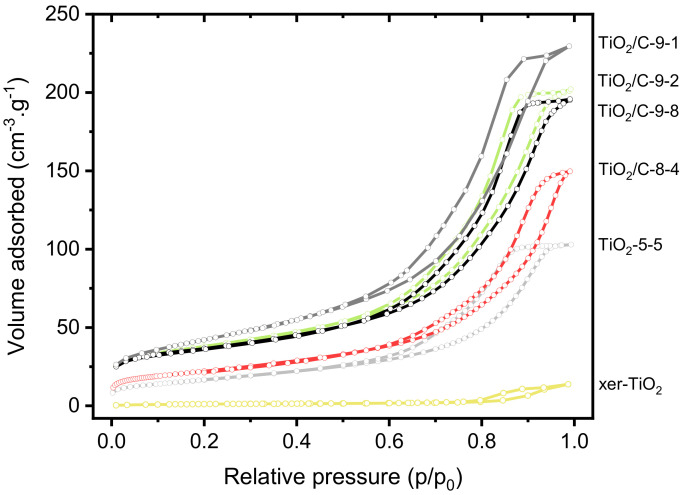
Nitrogen sorption isotherms at 77 K of xer-TiO_2_, TiO_2_-5-5 and the different TiO_2_/C-*x*-*y* samples prepared herein.

**Figure 6 nanomaterials-10-01789-f006:**
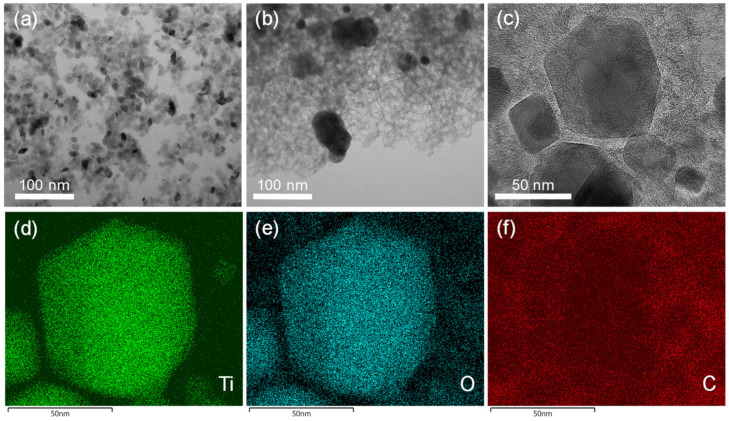
Transmission electron microscopy (TEM) images of (**a**) xer-TiO_2_ and (**b**) TiO_2_/C-9-1. (**c**) High resolution TEM image and (**d**–**f**) STEM-EDX mapping of TiO_2_/C-9-1. The corresponding element (Ti, O or C) is noted in the bottom left corner of each image.

**Figure 7 nanomaterials-10-01789-f007:**
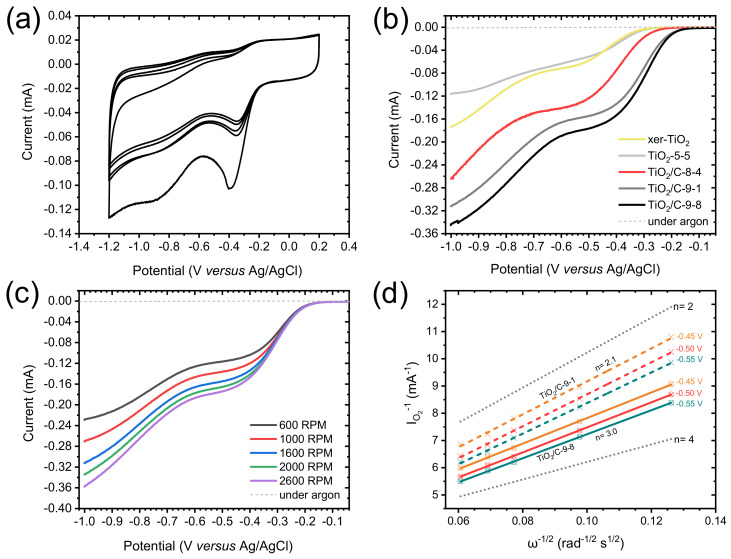
(**a**) Cyclic voltammogram measured for TiO_2_/C-9-1 in O_2_-saturated 0.1 M KOH solution at a scan rate of 100 mV s^−1^. (**b**) Polarization curves measured in O_2_-saturated 0.1 M KOH solution, at a scan rate of 10 mV s^−1^, and a rotation rate of 1600 rpm for xer-TiO_2_, TiO_2_-5-5 and the TiO_2_/C-*x*-*y* series. (**c**) Polarization curves measured in O_2_-saturated 0.1 M KOH solution (solid lines) for TiO_2_/C-9-1 at different rotation rates from 600 to 2600 rpm, at a scan rate of 10 mV s^−1^. The dashed lines are the polarization curves measured in Ar-saturated 0.1 M KOH solution, at a scan rate of 10 mV s^−1^ and a rotation rate of 1600 rpm. (**d**) Koutecky–Levich plots obtained for TiO_2_/C-9-1 (colored dashed lines) and TiO_2_/C-9-8 (colored solid lines) at different voltages (from −0.45 to −0.55 V vs. Ag/AgCl). The dotted lines correspond to the ideal two electrons process (n = 2) and four electrons process (n = 4).

**Table 1 nanomaterials-10-01789-t001:** Chemical composition, textural and structural properties of xer-TiO_2_, TiO_2_-5-5 and the different TiO_2_/C-*x*-*y* samples.

Sample	Organic (wt%) ^†^	C (wt%) ^‡^	H (wt%) ^‡^	SSA_BET_ (m^2^ g^−1^) ^#^	V_0.99_ (cm^3^ g^−1^)	TiO_2_ Phase(s)	Ti*_n_*O_2*n*−1_ Phases
xer-TiO_2_	31 ± 3	20 ± 2	2.4 ± 0.2	4	0.02	Anatase (A)	None
TiO_2_-5-5	n/a	n/a	n/a	60	0.16	A	None
TiO_2_/C-8-4	4.0 ± 0.4	5.0 ± 0.1	0.1	79	0.23	A/Rutile (R)	None
TiO_2_/C-9-1	6.8 ± 0.6	6.3 ± 0.1	< 0.1	150	0.35	A	Ti_3_O_5_/Ti_6_O_11_
TiO_2_/C-9-2	6.2 ± 0.5	n/a	n/a	130	0.31	A	Ti_3_O_5_/Ti_6_O_11_
TiO_2_/C-9-8	6.2 ± 0.5	6.3 ± 0.1	< 0.1	124	0.30	A	Ti_3_O_5_/Ti_6_O_11_

^†^ TGA weight loss from 150 to 800 °C; ^‡^ Elemental analysis; **^#^** BET equivalent specific surface area; Total pore volume determined at P/P_0_ = 0.99.

**Table 2 nanomaterials-10-01789-t002:** Electrochemical properties towards the ORR in 0.1 M KOH of xer-TiO_2_, TiO_2_-5-5 and the different TiO_2_/C-*x*-*y* samples studied herein. Comparison with a commercial Pt/C, 20 wt.%, catalyst.

Sample	Onset potential (V) ^a^	E_cat/2_ (V) ^b^	I_cat_ (mA mg^−1^) ^c^	I_k_ (mA mg^−1^) ^d^	n_electrons_ ^e^
xer-TiO_2_	−0.20	−0.43	1.27	n/a	n/a
TiO_2_-5-5	−0.20	−0.40	1.44	n/a	n/a
TiO_2_/C-8-4	−0.16	−0.39	2.80	n/a	n/a
TiO_2_/C-9-1	−0.09	−0.31	3.32	5.1	2.1 ± 0.2
TiO_2_/C-9-8	−0.09	−0.30	3.78	5.3	3.0 ± 0.2
Pt/C, 20 wt.%	−0.01	−0.13	4.80	10.0	3.8 ± 0.1

^a^ Onset potential determined at the base of the catalytic wave. ^b^ E_cat/2_ is the potential determined at half of the catalytic current according to Appel et al. [57]. ^c^ I_cat_ is the catalytic current determined at the first plateau region in the potential range from −0.4 to −0.7 V. ^d^ Kinetic current extracted from Koutecky–Levich (KL) plots at −0.4 V vs. Ag/AgCl. ^e^ Average electron transfer number extracted from KL plots applied from −0.35 to −0.55 V vs. Ag/AgCl.

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
