# Peer review of "TinO2n−1 Suboxide Phases in TiO2/C Nanocomposites Engineered by Non-hydrolytic Sol–Gel with Enhanced Electrocatalytic Properties"

_nanomaterials, 2020, doi:10.3390/nano10091789_

Round 1
Reviewer 1 Report
This paper reports on the synthesis of TinO2n-1/TiO2/carbon composites via the non-hydrolytic sol-gel (NHSG) process. The authors discuss the carbothermal reduction of TiO2 based on various analyses. The authors also investigated the electrocatalytic activity towards the oxygen reduction reaction (ORR) and reveal the superior capability of TinO2n-1 compared with TiO2 and carbon.
The reviewer thinks that this work has a potential to attract a broad interest in the fields of material synthesis and electrochemistry. After the revision as pointed out below, this paper is to be acceptable for publication.
- The survival of a substantial amount of anatase after the calcination at as high as 950 ºC is extraordinary. According to the thermodynamics, the crystal transition from anatase to rutile should take place in this temperature range. The authors should investigate more detail and give a discussion. The reviewer deduces that involvement of Cl- in TiO2 possibly suppresses the crystal transition.
- The authors had better investigate the carbothermal reduction behaviors at higher temperatures in this system in order to get more insights.
- The authors attribute the yield of mixed phases of Ti6O11 and Ti3O5 irrespective of the duration time to the highly condensed and well crystallized anatase crystal prepared via the NHSG process. However, even starting from TiO2 prepared by a normal sol-gel route, the calcined at high temperature enhances the crystallinity of anatase probably higher than the as-prepared anatase by NHSG. The reviewer thinks that the non-equilibrium reaction reflected by the mixture of TiO2/Ti6O11/Ti3O5 might be due to the synthesis condition. Calcining a compacted pellet prepared from grounded powders of the xerogels would give a different result.
- There is a typo in l.142, p.4. The “showed” should be “shown”.
Author Response
Dear Editor, Dear Reviewers,
We are grateful to referees for their careful review and valuable comments. Please see enclosed our response to each comment.
Reviewer 1
This paper reports on the synthesis of TinO2n-1/TiO2/carbon composites via the non-hydrolytic sol-gel (NHSG) process. The authors discuss the carbothermal reduction of TiO2 based on various analyses. The authors also investigated the electrocatalytic activity towards the oxygen reduction reaction (ORR) and reveal the superior capability of TinO2n-1 compared with TiO2 and carbon.
The reviewer thinks that this work has a potential to attract a broad interest in the fields of material synthesis and electrochemistry. After the revision as pointed out below, this paper is to be acceptable for publication.
- The survival of a substantial amount of anatase after the calcination at as high as 950 ºC is extraordinary. According to the thermodynamics, the crystal transition from anatase to rutile should take place in this temperature range. The authors should investigate more detail and give a discussion. The reviewer deduces that involvement of Cl- in TiO2 possibly suppresses the crystal transition.
We agree that the presence of a substantial amount of anatase after pyrolysis at 950 ºC is remarkable. As reported in literature [Carbon coating of anatase-type TiO2 and photoactivity, J. Mater. Chem. 2002, 12, 1391-1396], the presence of carbon layers on coated titanium dioxyde nanoparticles can suppress the phase transformation of anatase to rutile structure by preventing sintering and crystal growth. The effect of carbon coating on crystal transition was also observed in our previous work dealing with similar TiO2/C nanocomposites prepared by non-hydrolytic sol-gel (NHSG) [Angel Manuel Escamilla-Pérez, PhD Thesis, Univ. Montpellier, 2017; Chem. Eur. J. 2018, 24, 4982-4990]. Following the procedure described in this manuscript, TiO2 xerogels were prepared from two different oxygen donors, i.e. diisopropyl ether (IPE) and tetrahydrofuran (THF). The carbon content in the xerogel was significantly higher starting form THF (c.a. 20 wt%) than from IPE (c.a. 6 wt%). The partial crystal transition from anatase to rutile was observed after pyrolysis at 900°C for 4 hours under argon, yielding TiO2/C-IPE and TiO2/C-THF. Interestingly, the crystal transition from anatase to rutile was less advanced for TiO2/C-THF than for TiO2/C-IPE (see XRD patterns below), suggesting that the carbon content (related to the thickness of the carbon coating layer) strongly affects the crystal transition kinetic.
A few sentences together with new references were added in the revised manuscript to discuss this point (please see below).
« The presence of a substantial amount of anatase (A) after pyrolysis at 850 °C and 950°C is remarkable, since the anatase to rutile crystal transition usually occurs below 700 °C [Spectroscopic Investigation of the Anatase-to-Rutile Transformation of Sol−Gel-Synthesized TiO2 Photocatalysts, J. Phys. Chem. C 2009, 113, 36, 16151–16157]. As reported in literature [Carbon coating of anatase-type TiO2 and photoactivity, J. Mater. Chem. 2002, 12, 1391-1396], the presence of carbon layers onto coated titanium dioxyde nanoparticles can suppress the anatase to rutile phase transformation by preventing sintering and crystal growth. In our previous works [Angel Manuel Escamilla-Pérez, PhD Thesis, Univ. Montpellier, 2017; Chem. Eur. J. 2018, 24, 4982-4990], the same feature was observed for TiO2/C nanocomposites prepared by a similar NHSG procedure and pyrolyzed under argon atmosphere at 750 and 900°C. Our results suggested that the carbon content – which was related to the thickness of the carbon layer onto TiO2 nanoparticles – strongly affects the anatase to rutile crystal transition [Angel Manuel Escamilla-Pérez, PhD Thesis, Univ. Montpellier, 2017]. »
- The authors had better investigate the carbothermal reduction behaviors at higher temperatures in this system in order to get more insights.
In this study, our aim was to explore the preparation of TinO2n-1/TiO2/carbon nanocomposites through NHSG and carbothermal reduction at a rather low temperature (i.e. 950°C). Such temperature was not only selected based on thermogravimetric analyses, but also based on previous studies reported in literature for the preparation of titanium suboxides via carbothermal reduction (mostly carried out between 900 and 1000°C). We agree that the investigation of the carbothermal reduction at higher temperatures might provide further insights (in the same way as using different oxygen donors or employing alternative reduction protocols). We believe, however, that such an investigation would deserve a full extra study to be published at a later stage.
- The authors attribute the yield of mixed phases of Ti6O11 and Ti3O5 irrespective of the duration time to the highly condensed and well crystallized anatase crystal prepared via the NHSG process. However, even starting from TiO2 prepared by a normal sol-gel route, the calcined at high temperature enhances the crystallinity of anatase probably higher than the as-prepared anatase by NHSG. The reviewer thinks that the non-equilibrium reaction reflected by the mixture of TiO2/Ti6O11/Ti3O5 might be due to the synthesis condition. Calcining a compacted pellet prepared from grounded powders of the xerogels would give a different result.
Unlike classical hydrolytic sol-gel routes, NHSG routes usually yield highly condensed and well crystallized metal oxide particles, right after solvothermal synthesis and without further calcination at higher temperature. Thus, we suggested in the conclusion that these native TiO2 anatase nanocrystals are more resistant to the carbothermal reduction. This is not a statement but an assumption based on our knowledge. Obviousely, we fully agree with the reviewer that other parameters might explain this feature, such as the nature and the reactivity of the carbon produced by the NHSG route (which is another assumption proposed in the conclusion - page 11). Finally, as proposed by the reviewer, the calcination of a compacted pellet should give a different result. We believe, however, that such an investigation would deserve a full extra study to be published at a later stage.
- There is a typo in l.142, p.4. The “showed” should be “shown”.
The typo was corrected accordingly.

Reviewer 2 Report
This is a nicely written paper which contains yseful information and describes important research. In my opinion it can be published in the present form.
Author Response
Reviewer 2
This is a nicely written paper which contains yseful information and describes important research. In my opinion it can be published in the present form.
The authors would like to thank Reviewer 2 for this highly positive comment.
Reviewer 3 Report
The manuscript reports application of solvothermal treatment to production of reduced titania phases/carbon nanocomposites. The work has been carried out with care, but the technique is apparently not very efficient in this particular application. The reduction appears incomplete.
The path proposed in Fig. 1 is most probably correct, but lacks direct confirmation - high resolution TEM is absolutely necessary here to both prove the nature of the reaction product and also to explain how the incomplete reduction proceeds.
Why was THF used as reactant? - it is a volatile solvent hazardous for human health and easily forming explosive peroxides.
What was the interest in oxygen reduction reaction? What were the expected products? Some discussion of potential applications would be suitable.
Author Response
Reviewer 3
The manuscript reports application of solvothermal treatment to production of reduced titania phases/carbon nanocomposites. The work has been carried out with care, but the technique is apparently not very efficient in this particular application. The reduction appears incomplete.
The path proposed in Fig. 1 is most probably correct, but lacks direct confirmation - high resolution TEM is absolutely necessary here to both prove the nature of the reaction product and also to explain how the incomplete reduction proceeds.
We do not agree with reviewer 3 as XRD gave an unambiguous and direct confirmation to prove the nature of the reaction products, i.e. TiO2 (anatase and rutile) together with Ti6O11 and Ti3O5. As mentioned by reviewer 3, high resolution TEM might give precious insights to explain how the incomplete reduction proceeds. As you can see below, we performed high resolution TEM on TiO2/C-9-1. Crystal planes were clearly identified, confirming that nanocrystals were obtained. However, we could only identify what can be attributed to the (101) crystal planes of anatase. Unfortunately, we couldn’t get higher resolution.
Why was THF used as reactant? - it is a volatile solvent hazardous for human health and easily forming explosive peroxides.
Organic ethers are commonly employed as oxygen donors in NHSG for the preparation of TiO2 nanocrystals starting from TiCl4. In a previous study, we proposed to use two different oxygen donors, i.e. diisopropyl ether (IPE) and tetrahydrofuran (THF) to prepare TiO2 xerogels and TiO2/C nanocomposites [Chem. Eur. J. 2018, 24, 4982-4990]. The carbon content in the xerogel was significantly higher starting form THF (c.a. 20 wt%) than from IPE (c.a. 6 wt%). In the case of THF, the carbon content corresponds to a C:Ti molar ratio of 1.9 (vs. 0.6 in the case of IPE), that should be theoretically sufficient to fully reduce TiO2. For this reason, we selected THF as a model oxygen donor in this study. Besides, we agree that THF is flammable and hazardous for human health. Thus, innocuous alternatives must be selected to make the whole procedure safer and more sustainable. A sentence was added in the conclusion to address this point (see below).
“The use of innocuous alternatives to THF, such as polysaccharides [Dehydration of alginic acid cryogel by TiCl4 vapor: a direct access to mesoporous TiO2@C nanocomposites and their performance in lithium ion batteries, ChemSusChem, 2019, 12, 2660-2670], should be also investigated to make the whole procedure safer and more sustainable.”
What was the interest in oxygen reduction reaction? What were the expected products? Some discussion of potential applications would be suitable.
A few sentences and equations were added in the discussion (please see below). Five references were also added.
“The ORR is one of the most studied reaction in the field of electrochemistry, especially when dealing with energy storage and conversion devices [Metal-Free Catalysts for Oxygen Reduction Reaction, Chem. Rev. 2015, 115, 11, 4823–4892] [Recent Advances in Electrocatalysts for Oxygen Reduction Reaction, Chem. Rev. 2016, 116, 6, 3594-3657]. In particular, ORR in aqueous alkaline media [Oxygen Reduction in Alkaline Media: From Mechanisms to Recent Advances of Catalysts, ACS Catalysis 2015, 5, 4643-4667] has been widely studied and is of great interest for applications in metal-air batteries [Recent advances in zinc–air batteries, Chem. Soc. Rev., 2014, 43, 5257-5275] and alkaline anion-exchange membrane fuel cells [Alkaline Anion-Exchange Membrane Fuel Cells: Challenges in Electrocatalysis and Interfacial Charge Transfer, Chem. Rev. 2019, 119, 11945–11979]. In aqueous alkaline media, the ORR can proceed either by a direct four-electron pathway to directly reduce O2 into hydroxide ions (equation 1), or by a two-electron pathway with formation of peroxide ions as intermediate species (equation 2), followed by a two-electron reduction to hydroxide ions (equation 3) or disproportionation (equation 4).
”
